# Characterization of low-density granulocytes in COVID-19

Luz E. Cabrera[1]*, Pirkka T. Pekkarinen[2,3,4], Maria Alander[5], Kirsten H. A. Nowlan[2], Ngoc Anh Nguyen[2], Suvi Jokiranta[2], Suvi Kuivanen[1], Anu Patjas[6,7], Sointu Mero[6], Sari H. Pakkanen[6], Santtu Heinonen[8], Anu Kantele[6,7], Olli Vapalahti[1,9,10], Eliisa Kekäläinen[2,4,9], Tomas Strandin[1]

1 Zoonosis Unit, Department of Virology, University of Helsinki, Helsinki, Finland, 2 Department of Bacteriology and Immunology, University of Helsinki, Helsinki, Finland, 3 Division of Intensive Care Medicine, Department of Anaesthesiology, Intensive Care and Pain Medicine, University of Helsinki and Helsinki University Hospital, Helsinki, Finland, 4 Translational Immunology Research Program, Faculty of Medicine, University of Helsinki, Helsinki, Finland, 5 Department of Internal Medicine, University of Helsinki and Helsinki University Hospital, Helsinki, Finland, 6 Human Microbiome Research Program, Faculty of Medicine, University of Helsinki, Helsinki, Finland, 7 Inflammation Center, Division of Infectious Diseases, Helsinki University Hospital and University of Helsinki, Helsinki, Finland, 8 New Children's Hospital, Pediatric Research Center, Helsinki University Hospital and University of Helsinki, Helsinki, Finland, 9 Division of Virology and Immunology, HUSLAB Clinical Microbiology, HUS Diagnostic Center, Helsinki University Hospital, Helsinki, Finland, 10 Department of Veterinary Biosciences, University of Helsinki, Helsinki, Finland

* luz.cabreralara@helsinki.fi

**Data Availability Statement:** All relevant data are within the manuscript and its Supporting Information files.

**Funding:** This work was financed by grants by the Academy of Finland to T.S. (321809 and 328807),

## Abstract

Severe COVID-19 is characterized by extensive pulmonary complications, to which host immune responses are believed to play a role. As the major arm of innate immunity, neutrophils are one of the first cells recruited to the site of infection where their excessive activation can contribute to lung pathology. Low-density granulocytes (LDGs) are circulating neutrophils, whose numbers increase in some autoimmune diseases and cancer, but are poorly characterized in acute viral infections. Using flow cytometry, we detected a significant increase of LDGs in the blood of acute COVID-19 patients, compared to healthy controls. Based on their surface marker expression, COVID-19-related LDGs exhibit four different populations, which display distinctive stages of granulocytic development and most likely reflect emergency myelopoiesis. Moreover, COVID-19 LDGs show a link with an elevated recruitment and activation of neutrophils. Functional assays demonstrated the immunosuppressive capacities of these cells, which might contribute to impaired lymphocyte responses during acute disease. Taken together, our data confirms a significant granulocyte activation during COVID-19 and suggests that granulocytes of lower density play a role in disease progression.

## Author summary

The emergence of SARS-COV-2 and the ensuing COVID-19 disease has revealed an unprecedented need to understand the pathological mechanisms of acute respiratory infections in more detail. Granulocytes are highly abundant cells of the innate immunity,

O.V. (336490), S.H. (323499), A.K. (336439 and 335527); grants by the Helsinki University Hospital funds to P.T.P. (M7100YLIT2) and to O.V. (TYH 2018322); EU Horizon 2020 programme VEO (grant 874735) to O.V.; Finnish governmental subsidy for Health Science Research (TYH 2021315) to A.K. The funders had no role in study design, data collection and analysis, nor decision to publish, or preparation of the manuscript. Academy of Finland: https://www.aka.fi/en/ Helsinki University Hospital funds: https://studies.helsinki.fi/instructions/article/scholarships-and-grants EU Horizon 2020 programme VEO: https://www.veo-europe.eu/about-veo/funding

**Competing interests:** The authors have declared that no competing interests exist.

and thus first responders towards acute infections. However, their excessive activation can cause unwanted tissue damage and detrimental effects in humans. This study identifies a population of low-density granulocytes (LDGs) in COVID-19 patient samples, which has been poorly described in the context of acute infections so far. These cells were subclassified and found to be mainly of immature phenotypes. Further characterization revealed COVID-19 LDGs as a phenotypically diverse population with immunosuppressive characteristics, which seemed to be in line with an elevated recruitment and activation of granulocytes. Altogether, these findings suggest LDG may play a role in COVID-19 disease progression.

## Introduction

In December 2019, a new coronavirus disease (COVID-19), caused by a novel sarbecovirus, SARS-CoV-2, emerged in Wuhan, China and spread rapidly to the rest of the world and became a pandemic. Similarly to other coronaviruses that cause severe infections, namely SARS-CoV and MERS-CoV, the symptoms of COVID-19 include fever, nonproductive cough, dyspnea, myalgia and fatigue, with radiographic findings consistent with atypical pneumonia [1]. In addition, the clinical manifestations can develop towards acute respiratory distress syndrome (ARDS), with the possibility to evolve into a multiorgan dysfunction syndrome [2]. Interestingly, increased neutrophil recruitment is consistently observed in severe COVID-19 [1,3–7] and the neutrophil count positively correlates with disease severity, while lymphocyte numbers are depleted in patients with a poorer outcome [8]. Consequently, an increased neutrophil to lymphocyte ratio (NLR) can be present [5]. Moreover, the NLR is an independent mortality risk factor for hospitalized COVID-19 patients [3,9]. Along with this neutrophil count increase, their release of oxidant enzymes, microbicidal proteins, and chromatin in the form of neutrophil extracellular traps (NETs) is elevated [10–12]. Although released with the purpose of containing infections, NETs can potentially worsen the patient's inflammatory state and propagate microvascular thrombosis [13–15], both of which represent a major issue in COVID-19 pathology.

Low-density granulocytes (LDGs), regarded as a subset population of neutrophils in other diseases, are associated with an enhanced capacity to form NETs [16,17] and could contribute to COVID-19 pathophysiology. Unlike their normal-density counterparts, these cells isolate together with peripheral blood mononuclear cells (PBMC) during density gradient centrifugation. In addition to their pro-inflammatory capacity, they also exhibit an immunosuppressive phenotype in certain autoimmune diseases and cancer. In fact, a considerable part of LDGs can consist of immunosuppressive granulocytic myeloid-derived suppressor cells (G-MDSC) [18] and, compellingly, a massive expansion of MDSCs during COVID-19 has been demonstrated as a hallmark of the disease [19,20]. However, little is known concerning LDGs in infectious diseases in general [21]. Therefore, the present study intends to shed light on the role of these cells in COVID-19.

## Results

### The frequency of LDGs is increased in acute COVID-19

To analyze LDG frequencies in mild (outpatients) and severe (hospitalized) COVID-19 patients (1–3 weeks after onset of disease), as well as age- and sex-matched healthy controls (HC) (see S1 Table for patient characteristics), the PBMC fraction was isolated from fresh peripheral blood, stained with a panel of antibodies detecting specific granulocyte maturation

and activation levels (panels described in S2 Table) and analyzed by flow cytometry. UMAP and FlowSOM clustering analysis of live, CD3$^-$/CD14$^-$/CD19$^-$/CD56$^-$ cells (cleaned from lymphocyte and monocyte lineages) revealed mild and severe COVID-19 patients to harbor clearly distinct population distribution than HC (Fig 1A, UMAP of live PBMCs without lymphocyte and monocyte lineage removal depicted in S1 Fig). Based on the differential expression of granulocyte markers CD66, CD15, CD11b, CD16, CD33, and HLA-DR, 15 distinct clusters with variable densities between patient groups in live CD3$^-$/CD14$^-$/CD19$^-$/CD56$^-$ cells were identified (visualized in different colors in Fig 1A). Among the seemingly increased populations in COVID-19 patient samples, four were positive for both CD66 and CD15 and were therefore considered as low-density granulocytes (LDGs). The increased frequencies of LDGs in mild and severe COVID-19 as compared to HC was confirmed by conventional gating techniques of CD66 and CD15 co-expression in live CD3$^-$/CD14$^-$/CD19$^-$/CD56$^-$ populations (mild COVID-19, p < 0.01; severe COVID-19 p < 0.0001) (Fig 1C, representative gating in Fig 1B). The increased presence of granulocyte-like cells in PBMC of acute COVID-19 was evident also by an increased number of events in the high side scatter area, SSC-A (representative SSC-A vs. FSC-A plot in Fig 1D).

## Four distinct LDG subsets in variable stages of maturation are present in COVID-19

The four LDG subsets, identifiable by their co-expression of CD66 and CD15, differed in their levels of expression of granulocytic maturation markers CD33, CD16, and CD11b (cluster #1 CD33$^{++}$CD16$^-$CD11b$^-$, cluster #2 CD33$^+$CD16$^-$CD11b$^+$, cluster #3 CD33$^{low}$CD16$^+$CD11b$^+$, cluster #4 CD33$^-$CD16$^+$CD11b$^{-/low}$; heatmap of mean fluorescence intensity (MFI) of different markers in individual populations shown in Fig 2A). This expression pattern suggested clusters #1 to #3 to represent granulocytes of different developmental stages and cluster #4 to represent fully mature granulocytes. Accordingly, fluorescence imaging of isolated COVID-19 LDGs indicated that the majority displayed a non-segmented nuclear morphology and increased cytoplasmic MPO expression, both signs of granulocyte immaturity (Fig 2B).

To further assess the different developmental stages present in COVID-19 LDGs, clusters were sorted based on their differential CD16 and CD11b expression by flow cytometry, and subjected to fluorescence imaging as above (Fig 2C). Cluster #1 was identified as promyelocytes, with an abundant nucleus and small cytoplasm; cluster #2 ranged from myelocytes to metamyelocytes, displaying an increased cytoplasmic content and more bent nuclei; and cluster #3 was mainly represented by hyposegmented bands. Cluster #4, apparently the fully mature phenotype due to its lack of CD33 marker, was of lower frequency and could not be separated reliably from cluster #3. Nonetheless, granulocytes with a mature polymorphonuclear appearance were detected in isolated unsorted LDGs (Fig 2B), which were scarce and showed a lower MPO signal than the more immature phenotypes.

As for the population distribution among patient groups, severe COVID-19 was represented by the highest frequencies of clusters #1, #3 and #4, whereas the highest frequency of cluster #2 was exhibited by mild cases (Fig 2D). No distinct differences in marker MFIs between patient groups in different clusters were observed (surface marker expression heatmaps for mild and severe COVID-19 as well as HC shown in S2 Fig).

## The appearance of LDG subsets in acute COVID-19 is linked to elevated recruitment and activation of granulocytes

Next, we questioned whether the increase in LDG subset frequencies was related to more granulocytes being recruited to fight the infection. We assayed the levels of neutrophil chemotactic

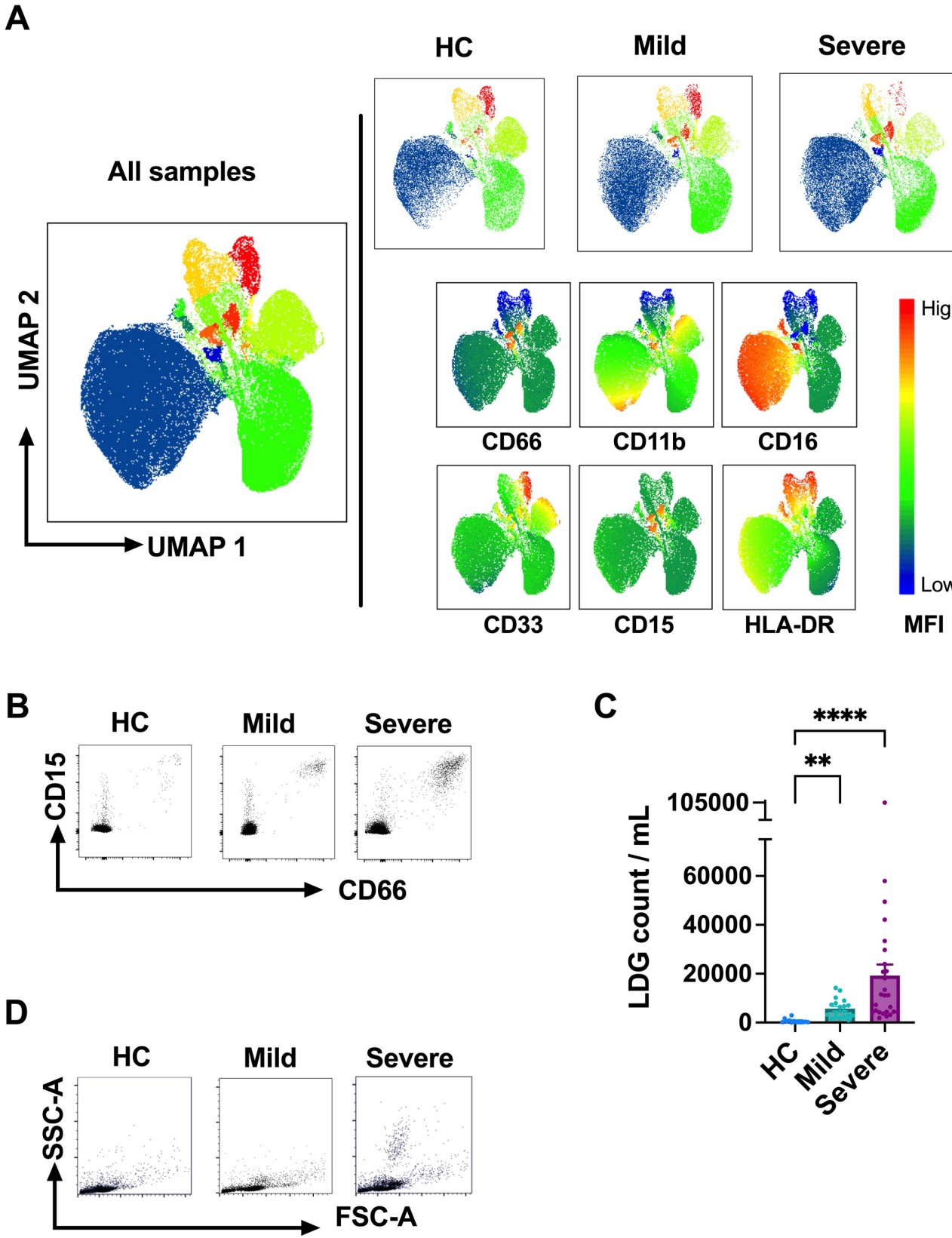

**Fig 1. The number of circulating LDGs increases in acute COVID-19.** (A) Fresh PBMCs obtained from acute mild (outpatients) and severe (hospitalized) COVID-19 as well as HC patients were stained with a panel of antibodies (see S2 Table for details) and analyzed by flow cytometry. Data from live CD3$^-$/CD14$^-$/CD19$^-$/CD56$^-$ cell populations were dimensionally reduced by Uniform Manifold Approximation and Projection (UMAP) and clustered by FlowSOM. (B) Representative identification of LDGs by flow cytometry in PBMCs of acute COVID-19 patients and HC. LDGs were identified as co-expressing CD66 and CD15 live CD3$^-$/CD14$^-$/CD19$^-$/CD56$^-$ cell populations. (C) The frequency of LDGs in PBMCs obtained from mild and severe COVID-19 patients (n = 21 and n = 34, respectively) during the acute stage (1–3 weeks after fever onset) vs. age- and sex-matched healthy controls (n = 14). To calculate circulating LDG counts per patient, the LDGs frequencies were normalized to lymphocyte counts, obtained by gating on their typical side and forward scatter characteristics in the whole PBMC fraction. From hospitalized (severe) COVID-19 patients, whole blood lymphocyte concentration was obtained from the white blood cell counts analyzed as part of the clinical care. In the case of HC and mild COVID-19 lymphocyte counts were assumed to be inside the normal range (2 million /ml blood). (D) Representative high side scatter (SSC-A) vs. forward side scatter (FSC-A) plots of PBMCs from COVID-19 patients and HC. Bars represent mean ± standard deviation. Statistically significant differences between groups are indicated by * p < 0.05, **** p < 0.0001.

factor IL-8 (CXCL8) and the main cytokines responsible for granulocytic precursor differentiation and granulopoiesis, granulocyte- and granulocyte/macrophage- colony stimulating factor (G-CSF and GM-CSF) in serum of mild and severe COVID-19 as well as HC. All cytokines were significantly elevated in the severe cases, versus mild COVID-19 and HC (Fig 3A–3C). All four LDG clusters in acute COVID-19 positively correlated with IL-8 levels (Fig 3G), with the increasingly mature clusters #3 and #4 showing an especially strong association (p < 0.0001 and p = 0.0004, respectively), whereas cluster #3 also positively correlated with G-CSF levels (p = 0.007).

To assess the extent of granulocyte activation and its association with LDG subset frequencies during acute COVID-19, we measured the circulating NETs, in the form of MPO-DNA complexes, together with calprotectin and MPO as additional markers of granulocyte activation in the patients' sera. Again, the severe cases showed significantly higher levels of MPO-DNA complexes, calprotectin, and MPO, as compared to both HC and mild cases (Fig 3D–3F), wherein MPO-DNA complexes significantly correlated with cluster #3, whereas the association of calprotectin with cluster #4 was borderline significant (p = 0.082). These findings suggest that NETosis, but not necessarily granulocyte activation in general (e.g., degranulation), are strongly linked to the appearance of LDGs in severe COVID-19.

Moreover, clusters #3 and #4 correlated significantly with the level of oxygen supplementation, IL-6, and length of stay in a tertiary hospital (Fig 3G), all indicators of disease severity, which further suggests a link between CD16$^+$ LDG subsets and increased disease severity in acute COVID-19.

## COVID-19 LDGs are immunosuppressive

Lymphopenia is a common finding in COVID-19 and correlates with a more severe outcome. How SARS-CoV-2 causes lymphopenia is still unknown [18,20]. LDGs are known to suppress T cell proliferation in several other diseases [16,17]. This prompted us to test the immunosuppressive ability of severe COVID-19 LDGs *ex vivo*, by activating healthy donor T cells in the presence of supernatants obtained from a 24-h culture of isolated severe COVID-19 CD66$^+$ LDGs. Notably, T cells showed a diminished proliferation index in the presence of supernatants from individual acute COVID-19 CD66$^+$ LDGs, compared to HC CD66$^+$ PMNs (Fig 4A and 4B, proliferation without added supernatants was set to 100%). Due to these significant immunosuppressive effects, acute COVID-19 LDGs could be functionally considered as G-MDSCs.

A follow-up patient cohort was also included in the present study, which consisted of two patients that remained in ICU for an extended time (approx. 1 month) and exhibited sustained elevated levels of circulating LDGs (primarily of clusters #1 and #2, the promyelocyte/metamyelocyte types) throughout the hospitalization period (S3 Fig). To assess the

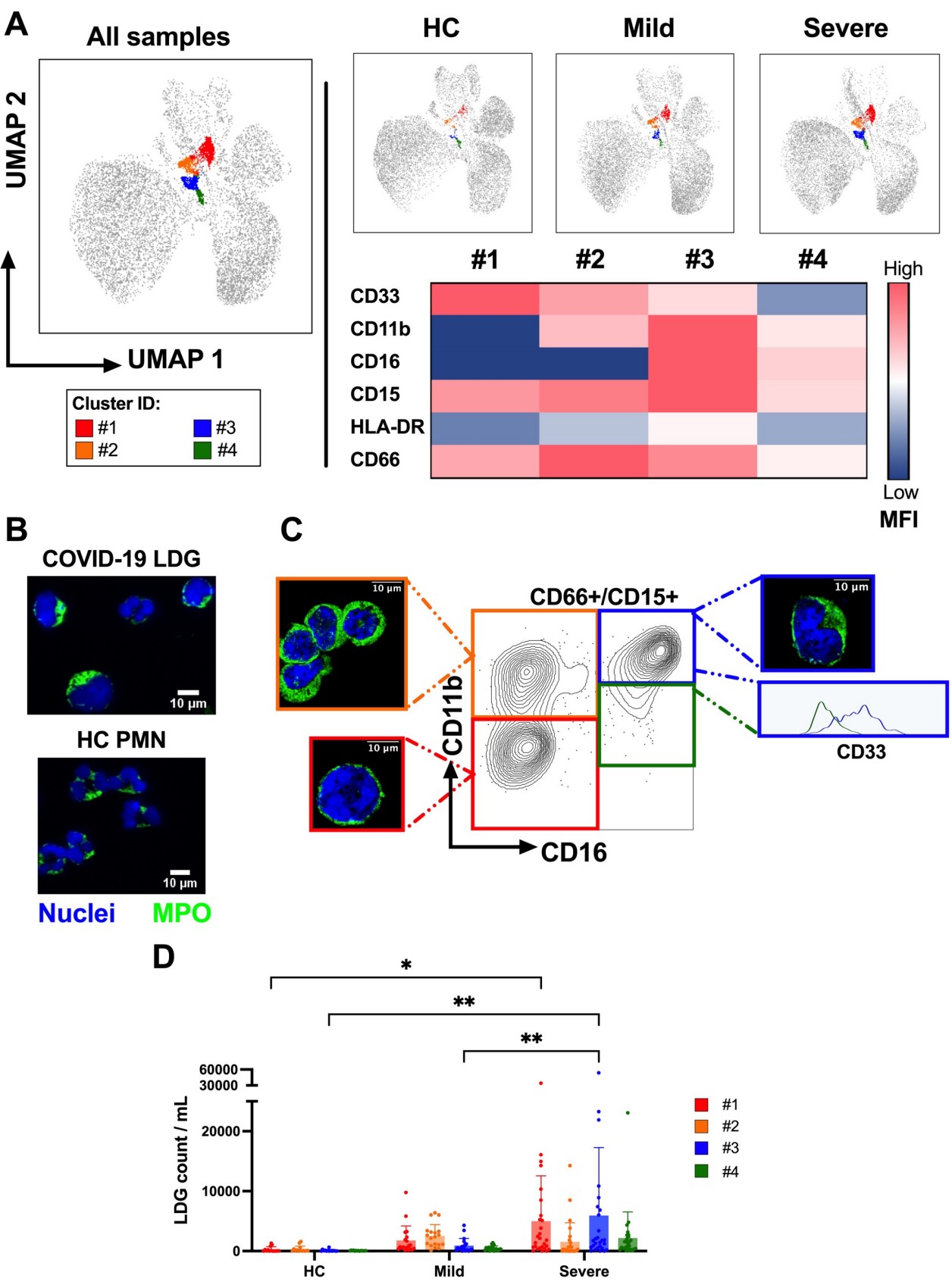

**Fig 2. Differential surface marker expression reveals immature COVID-19 LDG subsets.** (A) The CD66$^+$CD15$^+$ LDGs formed four different clusters with variable density, identified by UMAP and FlowSOM analysis in HC, mild and severe COVID-19. A detailed heatmap based on median fluorescence intensity (MFI) of CD33, CD11b, CD16, CD15, HLA-DR and CD66 in each cluster of severe COVID-19 patients is shown. (B) Fluorescence imaging analysis of nuclear morphology (Hoechst DNA stain) and cytoplasmic MPO expression of CD66+ LDGs and CD66+ PMNs, isolated by magnetic beads from severe COVID-19 PBMC and HC PMN fractions, respectively. (C) CD11b vs. CD16 plot of CD66$^+$CD15$^+$ LDGs using concatenated data from all patients, allowing separation of clusters #1 (CD16$^-$CD11b$^-$, red), #2 (CD16$^-$CD11b$^+$, orange), #3 (CD16$^+$CD11b$^+$, blue) and #4 (CD16$^+$CD11b$^{-/low}$, green). LDGs corresponding to clusters #1, #2 and #3 were flow cytometry-sorted based on their CD16 and CD11b expression and analyzed by fluorescence imaging as in (B). Histogram reveals differential CD33 expression in the CD16$^+$CD11b$^+$ quadrant, indicating overlap between clusters #3 and #4. (D) Comparison of the frequencies of each cluster per mL of blood in mild (n = 21) and severe (n = 34) COVID-19 patients as well as HC (n = 14), calculated similarly as in Fig 1. Statistically significant differences between groups are indicated by $^*$ p < 0.05, $^{**}$ p < 0.001.

immunosuppressive capacity of the LDGs throughout different phases of the disease, supernatants of the convalescent phase from these individual follow-up samples were studied in a similar manner than previously described for the acute stage of the disease. However, as depicted in the histogram from Fig 4A, no considerable inhibition effect by neither CD66$^+$ LDG nor PMN was seen during disease convalescence, in contrast with the acute phase. This was confirmed with pooled LDG or PMN supernatants indicating that only acute phase samples significantly inhibited T-cell proliferation (Fig 4C). This difference in the capacity to inhibit lymphocyte proliferation might be explained by the differential frequencies of LDG subpopulations between acute and convalescent patients, suggesting the more mature LDG phenotypes (clusters #3 and #4) are responsible for the observed immunosuppression.

## Increased frequency of LOX-1+ LDGs in severe COVID-19

Lectin-type oxidized LDL receptor-1 (LOX-1) was previously identified as a specific marker of G-MDSC in cancer patients [22]. To understand the potential immunosuppressive function of distinct LDG subsets in acute COVID-19, we analyzed the surface expression level of LOX-1 in the different LDG clusters of mild and severe COVID-19 patients as well as HC. Generally, the frequency of LOX-1$^+$ LDGs varied extensively between subpopulations in the order #2 > #3 > #1 > #4 (Fig 5A), with both mild and severe COVID-19 showing increased numbers, as compared to HC. Particularly, LDG cluster #4 of severe COVID-19 patients had significantly more LOX-1+ cells as compared to mild cases ($^*$ p < 0,05) (Fig 5A). These results suggest that severe COVID-19 LDGs could be more immunosuppressive than those of mild COVID-19 or HC (and LDGs of mild COVID-19 more suppressive than HC). Unfortunately, we were unable to directly assay the *ex vivo* immunosuppressive effect of LDGs of mild COVID-19 or HC due to their low frequencies in contrast with the severe cases.

We further hypothesized whether the variable LOX-1 positivity in acute COVID-19 LDGs could be due to differences in cellular activation level. To assess the role of LOX-1 in activation of LDGs, whole blood from healthy donors was stimulated *in vitro* with two well-known granulocyte activators: lipopolysaccharide (LPS) and phorbol-12-myristate-13-acetate (PMA). Both stimulators generated a significantly increased frequency of LOX-1$^+$ LDGs, while also increasing the total number of LDGs (Fig 5B). These results propose that inflammatory mediators can increment the frequency of LOX-1$^+$ LDGs by activating mature PMNs, which is a likely scenario also during acute COVID-19.

## Variable granulocyte subset distribution between LDGs and PMNs

Our findings of distinct mature LDG subsets during acute COVID-19, as well as the ability to generate LDGs from PMNs by *in vitro* activation (Fig 5B), urged us to speculate whether some of the identified LDG clusters in acute COVID-19 could be derived from "normal-density" PMNs in response to excessive granulocyte activation. Thus, isolated PMNs from peripheral

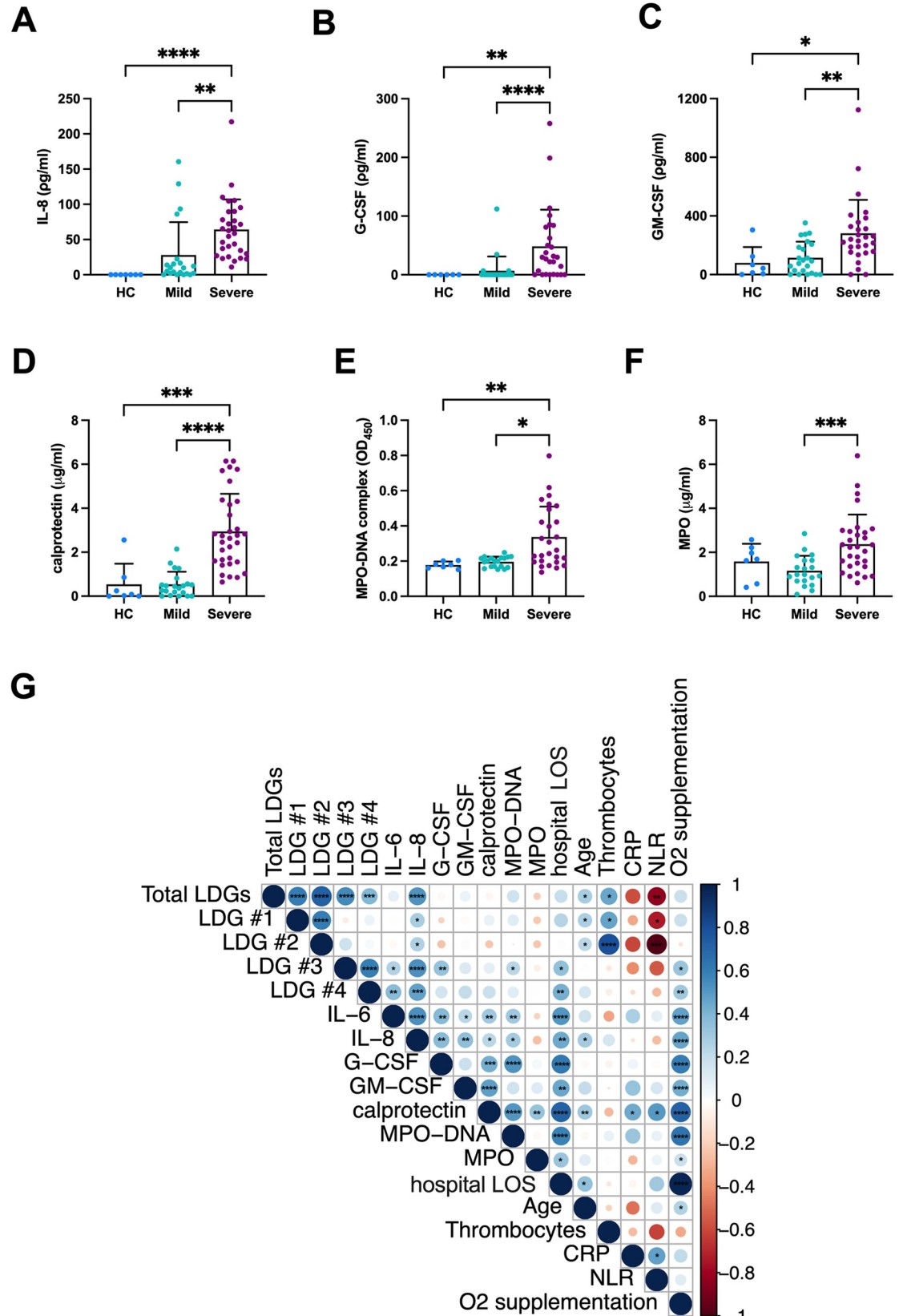

**Fig 3. The increased amounts of COVID-19 LDG subsets are linked to elevated recruitment and activation of granulocytes.**
(A-F) The serum concentrations of IL-8, G-CSF, GM-CSF, calprotectin, MPO-DNA complexes and MPO in mild and severe acute
COVID-19 as well as HC were measured by enzyme-linked immunosorbent assays. The dates of serum sampling matched those of

LDG measurements, shown in Figs 1 and 2. OD$_{450}$ = Optical density at 450 nm. Bars represent mean ± standard deviation. (G) A bivariate correlogram of LDG subset counts to clinical parameters and different factors measured in serum (those shown in A-F as well as IL-6) in severe COVID-19. CRP = C-reactive protein, NLR = neutrophil to lymphocyte ratio in whole blood, hospital LOS = Length of stay in tertiary hospital. The color and increasing size of the circles indicate spearman rank correlation coefficients and their p values, respectively. Statistically significant difference between groups (in A-H) and correlations (in G) were indicated by * $p < 0.05$, ** $p < 0.01$, *** $p < 0.001$ and **** $p < 0.0001$.

blood of both mild and severe acute COVID-19 were analyzed by flow cytometry, following the same strategy as with LDGs above. Unsupervised clustering methods demonstrated a different population composition between CD66$^+$CD15$^+$ LDGs and CD66$^+$CD15$^+$ PMNs during acute COVID-19, where PMNs mainly consisted of CD16$^+$/CD11b$^+$ (cluster #3) and CD16$^+$/CD11b$^{-/low}$ (cluster #4) (mild cases in Fig 6A and 6B and 6C and severe cases in S4A and S4B Fig; individual patients are shown in S4C and S4D Fig). No dramatic differences in population composition between patient groups were observed (Fig 6D), although severe COVID-19 PMNs displayed slightly increased frequency of potentially immature clusters #1 #2 as well as mature cluster #4. As expected, these results showed that the majority of PMNs both in acute COVID-19 and HC possess a mature phenotype, in contrast with the predominantly immature phenotypes of the isolated LDGs.

## Discussion

SARS-CoV-2 causes COVID-19 that can range from asymptomatic to severe or deadly, with patients presenting with acute respiratory symptoms that can potentially evolve towards respiratory failure [2]. The evidence suggests that immune cells play a major role in the pathophysiological mechanisms behind the patient-to-patient variable response to SARS-CoV-2 [7]; however, a detailed pathological role of different immune cell populations has yet to be elucidated. In the present study, granulocytes of lower density, known as LDGs, were targeted to characterize and determine their role in COVID-19.

Our findings revealed a significantly increased frequency of LDGs in acute COVID-19 patients' circulating blood, in comparison with age- and sex-matched healthy individuals, which was more prominent in the severe cases. This is congruent with recent biochemical and single-cell transcriptomic analyses revealing the presence of neutrophils in COVID-19 patient PBMCs [23–25]. Supporting earlier findings, COVID-19 LDGs showed immature nuclear morphologies, hereby detected through fluorescence imaging studies after undergoing a previously established sorting strategy of the different neutrophil developmental stages based on the expression of CD11b and CD16 [26,27]. Moreover, the 4 distinct subsets (cluster #1, #2, #3 and #4) identified by unsupervised methods based on their phenotypical surface marker expression represented the different maturation stages of granulocytes previously detected from whole blood [26,27]: promyelocytes; myelocytes, bands; and mature granulocytes. This was best exemplified by the expression of CD33, a granulocyte precursor marker that gradually decreased towards the more mature stages. These subpopulations presented a differential expression of CD16 and CD11b as well, which was not a surprise, given that these molecules are established markers used to identify granulocyte stages of differentiation [28–30]. A lower expression of CD16, suggestive of a lower degree of maturity [28], was indeed detected in the most immature subsets (cluster #1 and #2). CD11b expression, which also represents a continuum of increasing maturity, was absent in cluster #1 and high in clusters #2 and #3. However, it was decreased in the subpopulation #4, thus this cluster might represent an activated mature granulocytic phenotype, where the loss of CD11b expression can happen as a result of a strong stimulation and subsequent proteolytic degradation [31].

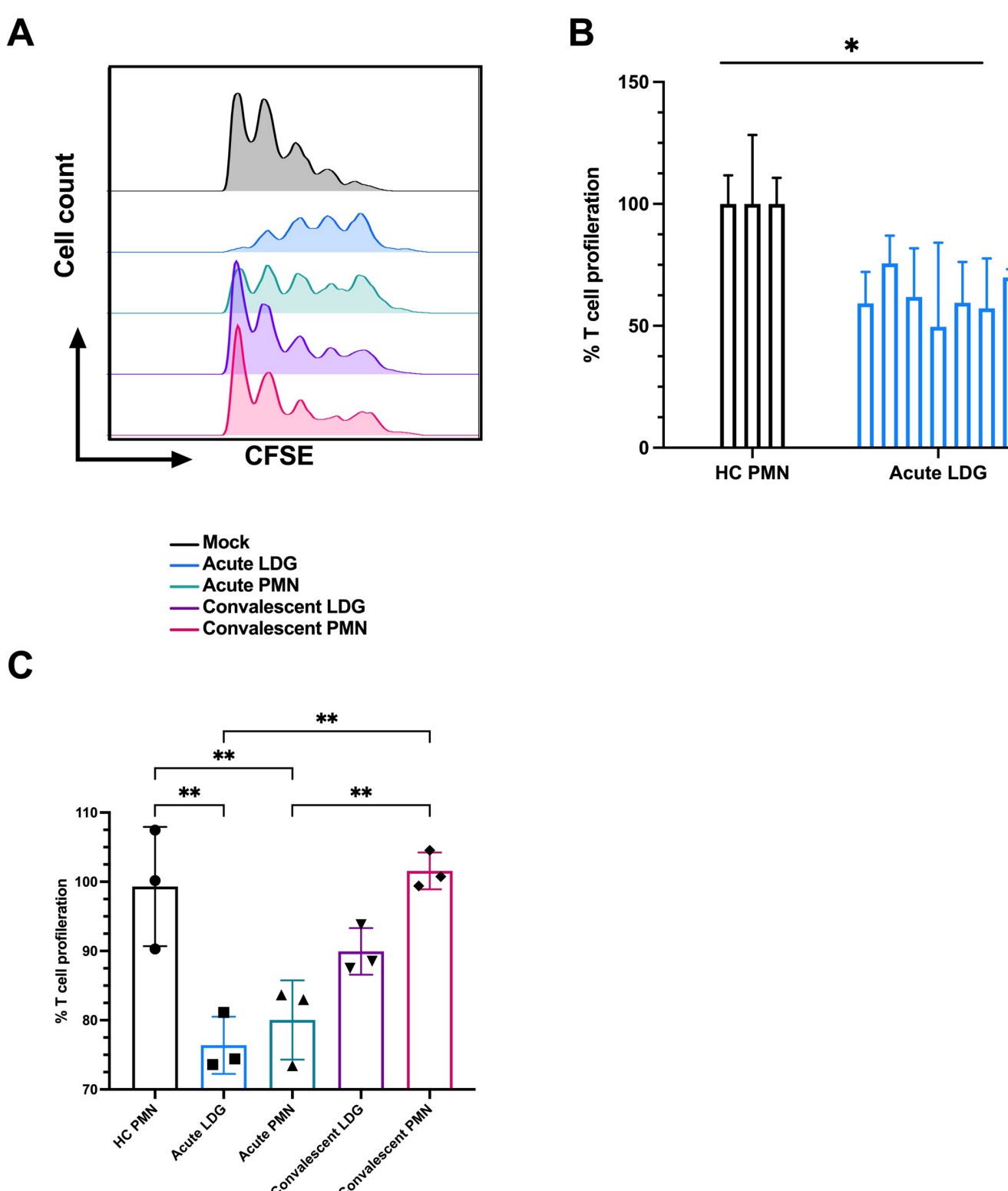

**Fig 4. COVID-19 LDGs are immunosuppressive.** (A) Representative CFSE histograms indicating the number of T cell generations produced by CD3/CD28 activation in the presence of supernatants from cultured acute COVID-19 LDGs and PMNs as compared to without supernatants (mock). (B) Percentage of T cell proliferation, as measured by proliferation index (PI), in the presence of supernatants from HC PMNs (n = 3) and COVID-19 LDGs (n = 7). (D) Percentage of T cell proliferation (n = 3), as measured by PI, in the presence of pooled supernatants of acute and convalescent COVID-19 LDGs and PMNs (n = 4 for acute, n = 2 for convalescent) as well as HC PMNs (n = 4). The PI in mock was set to 100%. Bars represent mean ± standard deviation. Statistically significant differences between groups were indicated by [*] $p < 0.05$, [**] $p < 0.01$, [***] $p < 0.001$.

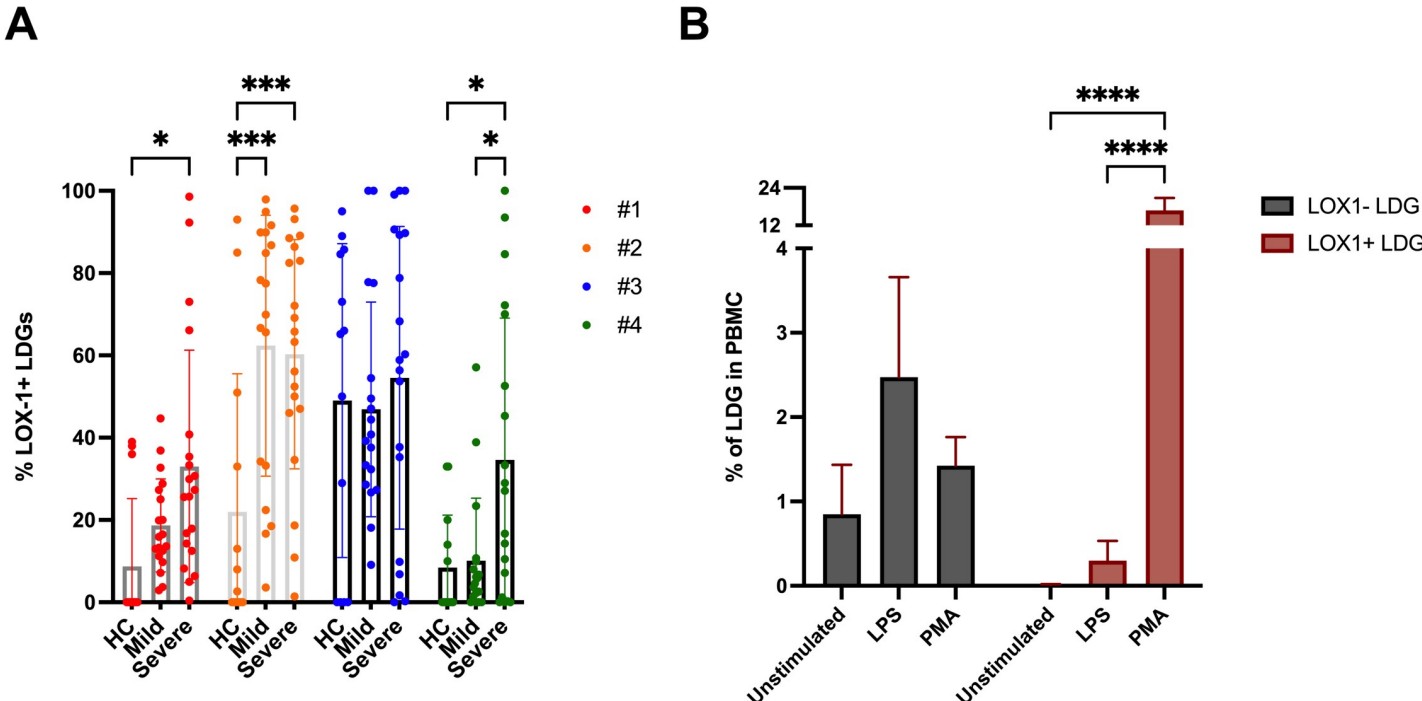

**Fig 5. LOX-1+ LDGs are increased in severe COVID-19.** (A) Percentage of LDG clusters #1-#4 expressing the LOX-1 surface marker in mild and severe COVID-19 as well as HC. (B) The percentage of LOX-1$^+$ and LOX-1$^-$ LDGs in PBMCs after 90min *in vitro* stimulation of whole blood with LPS or PMA (unstimulated as control). Statistically significant differences between groups indicated by $^*$ p < 0.05, $^{**}$ p < 0.01, $^{***}$ p < 0.001, $^{****}$ p < 0.0001.

The observed co-occurrence of LDG subpopulations at varying developmental stages is congruent with previous studies on LDGs in different diseases, in which they have been described as heterogeneous populations consisting of immature and mature "neutrophil-like" populations [18]. However, previous studies primarily describe LDGs in the context of a chronic conditions, whereas acute pathological infections are known to strongly activate innate immune responses. Although the responses involving granulocyte recruitment and activation at the site of infection are commonly described during infectious processes caused by bacteria, they can also occur during viral diseases [32–35]. When this happens, it can be followed by a replenishment of the lost granulocyte pools through emergency myelopoiesis [36], which is likely a physiological response to inflammation [37]. In fact, the increase of circulating immature cells, known in a clinical setting as a left shift, does not differ in COVID-19 from that of bacterial or other viral infections [38]. Since immature granulocytes typically display lower density than their mature counterparts [39], it is not surprising that a part of them would end up in the PBMC fraction during acute infectious processes. Accordingly, severe COVID-19 presented with significantly increased levels of IL-8, a neutrophil chemotactic factor [40], as well as G-CSF and GM-CSF, two myelopoietic cytokines [36,41]. Out of these factors IL-8 levels correlated with all LDG subsets, predominantly with the CD16$^+$ clusters #3 and #4, whereas G-CSF associated with cluster #3. These findings indicate that granulocyte recruitment and the subsequent increase in myelopoiesis likely contribute to the observed elevation of immature LDG frequencies in severe COVID-19. The immature clusters (#1 and #2) significantly correlated with the patients' thrombocyte counts in circulating blood, which goes in line with the emergency myelopoiesis, given that both cells share a common myeloid precursor [36,42,43]. In parallel, elevated numbers of megakaryocytes, the precursors of thrombocytes, have been demonstrated during COVID-19 [44].

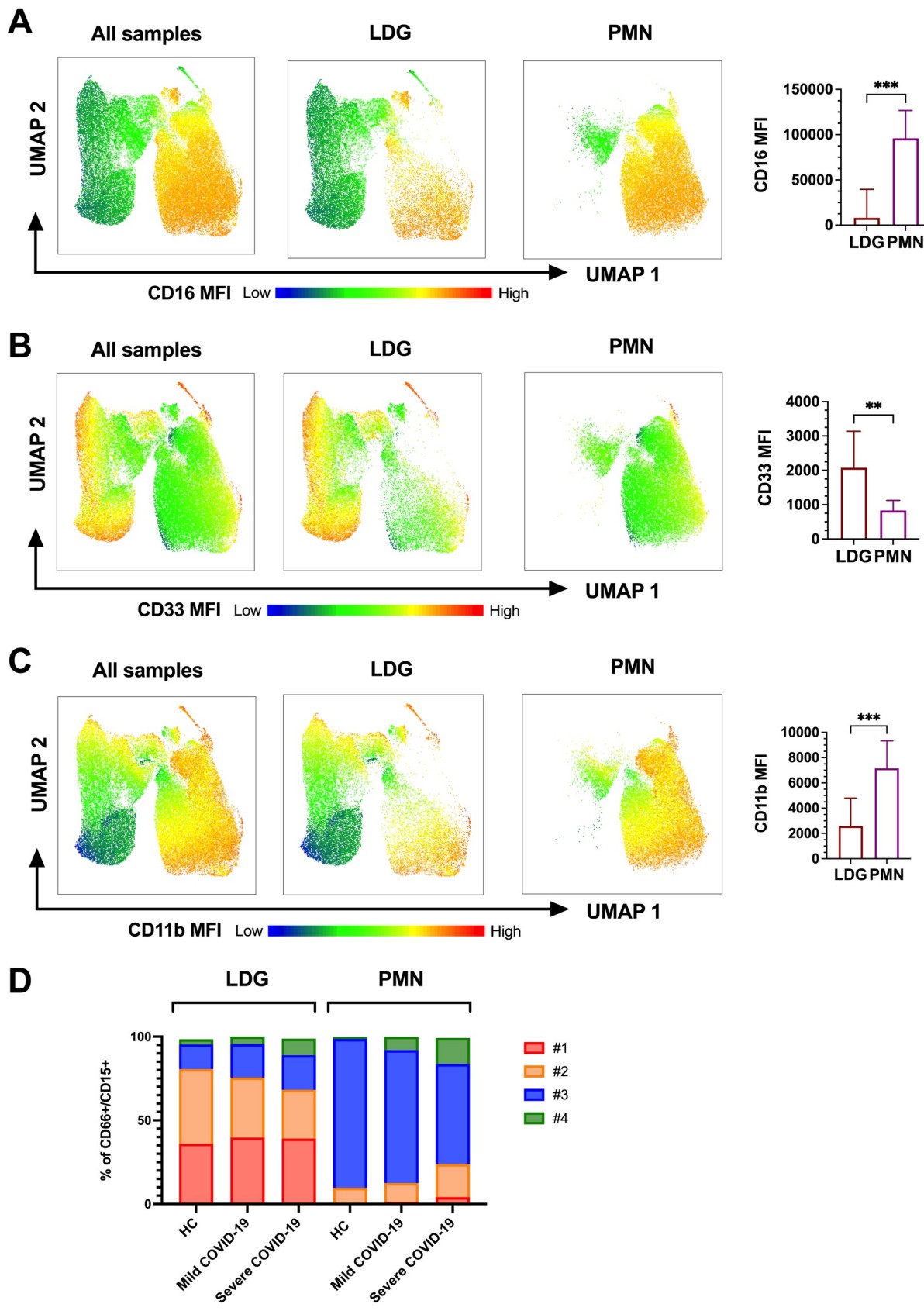

**Fig 6. Comparison between LDG subsets and normal density granulocytes (PMNs) in COVID-19.** Overlay of CD16 (A), CD33 (B) and CD11b (C) MFI heatmaps in dimensionally reduced LDG and PMN population density plots (left) and bar graphs of MFIs in individual patients (right). Dimensional reduction was performed by UMAP using CD15, CD66, CD11b, CD33 and HLA-DR MFIs of live CD3-/CD14-/CD19-/CD56- /CD66+/CD15+ population. (D) Mean frequencies of LDG subset clusters #1-#4 in LDGs and PMNs of mild and severe COVID-19 as well as HC.

In addition to neutrophil recruitment, an extensive activation of these cells has been described to occur during acute COVID-19 [45–47], though it is not clear whether this activation plays a role in the pathogenesis of COVID-19, or whether it reflects a normal response to other pathological processes occurring simultaneously, thus aggravating the clinical presentation. In fact, a recent study found no signs of hyperinflammation in the neutrophil compartment during COVID-19 at hospitalization [38].

In spite of their typical immaturity, granulocytes of lower density are known to possess an enhanced NETosis capacity [16,17], a process also elevated in COVID-19 [10,12]. In this study, circulating NETs were detected during the acute stage of the disease by the measurement of MPO-DNA complex assembly, which was significantly increased in severe COVID-19. Out of the LDG subsets, cluster #3 significantly associated with MPO-DNA, suggesting this subset to directly contribute to NETosis. In addition, clusters #3 and #4 significantly correlated with the level of oxygen supplementation, the proinflammatory cytokine IL-6 indicated in COVID-19 pathophysiology [48], and the length of stay in the hospital. In addition to the acute stage, the evolution of LDG populations during disease convalescence (weeks 4 to 5 post onset of disease) was also characterized. The two follow-up patients in our cohort displayed strongly increased frequencies of CD16- LDGs (clusters #1 and #2) during their prolonged hospitalization, which is similar to the LDG subset distribution observed in mild COVID-19. This goes well together with the idea that CD16+ LDGs are linked to COVID-19 disease severity, whereas the sole presence CD16- LDGs is predictive of recovery. These findings point towards the appearance of CD16+ LDG subsets in severe COVID-19 to be directly linked to disease progression and may even predict the need of respiratory support and length of hospitalization.

Given that: (a) LDGs can be a product of activation and degranulation of mature normal-density granulocytes with immunosuppressive capacity [18,49,50]; and (b) the granulocyte subsets emerging in severe COVID-19 express genes associated with immunosuppressive features [20]; the possibility of this emergence of LDGs associating with the lymphocyte depletion seen in COVID-19 was analyzed [8]. Indeed, we observed a marked T cell proliferation suppressive capacity by severe acute COVID-19 LDGs *in vitro*, which was not displayed by LDGs from the convalescent phase. This difference between acute and convalescent LDGs could be explained by their differing subset composition: the significant amounts of CD16+ LDGs detected during the acute disease, which associated with disease severity markers, were almost undetectable during the convalescence stage, similarly to mild COVID-19 and healthy conditions. Taken together, these results suggest that mature CD16+ LDGs are more immunosuppressive than immature CD16- LDGs, and the presence of the former is most likely due to excessive granulocyte recruitment and activation during acute, severe COVID-19.

These results confirm the immunosuppressive phenotype of COVID-19 LDGs and propose that these cells can function as G-MDSCs, also previously hypothesized [20] and recently also observed elsewhere [25]. Furthermore, the finding that immunosuppressive effects were observed with PMNs from acute disease phase also proposes granulocyte activation to be a prerequisite for the MDSC phenotype to occur, as implied elsewhere [50]. Interestingly, cancer-associated G-MDSCs have a CD16+/CD11b+ phenotype [51], which resembles cluster #3 in our study. LOX-1 has been proposed as an additional marker of G-MDSCs in cancer [22],

which prompted us to investigate its expression in the different LDG subsets. A major proportion of each subset was found LOX-1[+], which seems to further increase in acute COVID-19 as compared to HC. Furthermore, the proportion of LOX-1[+] LDGs in cluster #4 was significantly elevated in the severe cases, as compared to mild COVID-19. Our results are thus in line with the idea of LOX-1 as a marker of G-MDSCs, which is distinctly elevated during the acute stage of severe COVID-19. Unfortunately, we were unable to assess the immunosuppressive capacity of HC and mild COVID-19 LDGs, due to their relatively low frequencies, which consisted of mainly immature phenotypes. However, the extent of LOX-1[+] LDGs in mild cases suggests they could possess some immunosuppressive function, albeit at a lower level than in severe disease.

As mentioned above, LDGs could arise due to activation-induced cell density decrease of normal-density granulocytes [18,21,50]. To shed light on the surface marker expression and activation status of circulating normal-density granulocytes, the distribution of the identified cluster phenotypes present in LDGs were assessed in PMNs isolated from acute COVID-19 and healthy controls. In both cases, CD66[+]CD15[+] PMNs consisted mainly of the CD16[+] phenotypes (LDG clusters #3 and #4), which in this case lacked CD33 surface marker expression. The frequency of clusters #1, #2 and #4 in these PMNs was increased during severe COVID-19, relatively to HC and mild disease. Thus, CD16[+]/CD11b[+] (corresponding to cluster #3) no longer constituted 88% of the CD66[+]/CD15[+] in circulating whole blood, whereas cluster #4 increased from 1.3% to 15.6%. These changes suggest cluster #3 to represent mature non-activated PMNs, present in healthy and diseased, whereas cluster #4 is likely an activated phenotype with diminished CD11b surface expression, similar to LDG cluster #4 but without a sufficient loss of density to end up in the LDG fraction. Similarly to LDGs, the CD16[-] PMN (LDG clusters #1 and #2) are most likely immature granulocytes, exceptionally detected in healthy PMNs, but presenting a relatively increased frequency when isolating PMNs from COVID-19 patients, due to an overall elevation of the immature populations in the circulating peripheral blood. Taken together, these findings support the idea that during acute COVID-19, both emergency myelopoiesis and PMN activation contribute to the appearance of increased numbers of LDGs in the circulation, which can contribute to disease progression by amplification of pathological inflammatory processes, but also by modulating and suppressing the adequate adaptive immune responses.

In conclusion, part of the cellular immune response during COVID-19 is described in this study, concerning one of the first cellular responders to an infection: the granulocytes. These results revealed a remarkable expansion of the granulocytes present in the PBMC fraction during acute COVID-19, consisting of four distinct granulocyte subpopulations. We believe this increase of mainly immature subsets signifies emergency myelopoiesis initiated as a response to strong granulocyte recruitment and activation, as expected from an infectious inflammatory entity such as COVID-19. Our findings suggest that LDGs from acute severe COVID-19 differed significantly from LDGs present in mild disease or in health. They also differ from their normal density counterparts, both in cell subset composition and marker expression intensity. However, the G-MDSC function can be attributed to both LDGs and granulocytes of normal density in acute COVID-19. In brief, the expansion of granulocyte populations needs to be further investigated and characterized to define a possible role for these innate immunity cells as markers of disease activity, as well as potential therapeutic approaches targeting their detrimental effects during an acute infection.

## Materials & methods

### Ethics statement

The study was approved by the Ethics Committee of the Hospital District of Helsinki and Uusimaa (HUS/853/2020, HUS/1238/2020). All subjects gave a written informed consent in accordance to the Declaration of Helsinki.

### Patient population

Patients with confirmed COVID-19 (RT-PCR positive for SARS-CoV-2) at Helsinki University Hospital (HUH) (hospitalized n = 34 and outpatients n = 21) were included in the present study. As controls, age- and sex-matched healthy blood donors (n = 14) were included. The clinical presentation was variable among the patients (S1 Table). For the present study, COVID-19 patients were categorized by severity based on their need for hospitalization and O2 supplementation. Disease severity of hospitalized patients was stratified based on the highest O2 supplementation required as follows: 0 = no oxygen therapy, 1 = oxygen by mask or nasal prongs $< 60\%$, 2 = oxygen by mask $\geq 60\%$ or respiratory support by non-invasive ventilation/high-flow oxygen therapy and 3 = intubated and mechanically ventilated. Analysis was performed from blood samples taken at the earliest possible time after hospitalization. In addition, two patients (namely patient #1 and patient #2) with an ICU length of stay (LOS) of 17 and 27 days, respectively, were sampled throughout the acute and convalescent phase, until discharge approximately 5 weeks later.

### Flow cytometry

PBMCs or PMNs were separated from EDTA-anticoagulated whole blood by density gradient centrifugation either with Ficoll-Paque Plus (GE Healthcare) or Polymorphprep (Axis-Shield), respectively, using standard procedures including red blood cell lysis with ACK lysis buffer (Lonza by Thermo Fisher). Typically, $1\text{-}2\text{x}10^6$ cells were incubated with live/dead Fixable Yellow Dead Cell Stain Kit (1:1000) (ThermoFisher), followed by addition of 1% fetal calf serum (FCS) and Fc receptor blocking reagent (Immunostep) for 5 min at RT. Subsequently, cells were surface stained at RT for 30 min in the dark with the titrated fluorescently labeled anti-human mouse monoclonal antibody cocktail (S2 Table), followed by centrifugation at 400 x g for 5 min at 22°C and fixing in 2% paraformaldehyde (PFA) for 15 min RT in the dark. Finally, cells were subjected to flow cytometric analysis with a three-laser (Blue/Red/Violet lasers) 14-color Fortessa LSRII cytometer (BD Biosciences). Data was acquired with BD FACSDiva version 8.0.1 (BD) software and further analysis was performed with the FlowJo software v10.0.7 (BD). A proper instrument daily performance check was ensured by running CS&T beads (BD Biosciences). PBMCs were selected by means of the morphology gate (FSC-A/SSC-A plot) and gated by the Height (FSC-H) to Width (FSC-A) ratios in the forward scatter to exclude doublets. Only live cells were included in the analysis, and LDGs were defined as $CD3^-/CD56^-/CD19^-/CD14^-/CD66^+/CD15^+$ cells.

The recorded datasets were normalized independently before concatenation. Then, the collected data was subjected to unsupervised dimensionality reduction and identification of cell populations, by Uniform Manifold Approximation and Projection for Dimension Reduction (UMAP) [52]. To generate UMAP plots, the minimum distance was set at 0.5 and the nearest neighbors' distance was set at 15, using a Euclidean vector space. Further identification and characterization of clusters was possible with the use of flow self-organizing map (FlowSOM) [53], with a number of meta-clusters set to 15.

## Isolation and culture of CD66+ cells

From PBMCs or PMNs isolated as described above, CD66 positive cells were separated using CD66abce MicroBeads Kit (Miltenyi Biotec) for positive selection with an MS column. After isolation, cells were either cytocentrifuged (CytoSpin 3, Shandon) onto a microscope slide or cultured (500,000 cells/ml) for 24 hours in growth medium RPMI-1640 supplemented with 10% inactivated FCS, 100 IU/mL Penicillin, 100 μg/mL Streptomycin, 2 mM L-glutamine (R10) at 37˚C, after which cells were removed at 400 x g for 5 min at 22˚C and their supernatant was collected and finally stored in– 70˚C.

## Cell sorting

From PBMCs isolated using Ficoll gradient and stained with the previously described panel, cells were analyzed with BD FACSAria Cell Sorting System and BD FACSDiva Software (BD Biosciences). After negative selection for CD3+/CD14+/CD56+/CD19+ cells, the remaining gated cells were selected based on their CD66+/CD15+ double positivity and sorted based on their expression of CD16 and CD11b surface markers [26,27]. Following this, cells were either cytocentrifuged onto a microscope slide or cultured, as described above.

## Immunofluorescence microscopy

Cytocentrifuged LDGs were fixed for 20 min with 2% PFA at RT and later stored in PBS at +4˚C. Cells were permeabilized by incubating for 5 min in permeabilization and blocking buffer (PBB) consisting of 3% BSA and 0.1% Triton X-100 in phosphate buffered saline (PBS). Cells were incubated with anti-MPO (Bio-Rad) diluted 1:1000 in PBB and visualized using Alexa-488 goat anti-mouse antibody (Invitrogen, ThermoFisher). Nuclei were stained with Hoescht 33420 (ThermoFisher). After washing in PBS, cells were mounted in Immu-Mount (Shandon, ThermoFisher) and fluorescence micrographs were obtained with PerkinElmer Opera Phenix spinning disk confocal microscope (Finnish Institute of Molecular Medicine, Helsinki, Finland) using 40x water immersion objective. The micrographs were processed with Fiji ImageJ 1.53f [54].

## T-cell activation assay

T cells from freshly isolated PBMCs of three healthy donors were enriched through negative selection, through depletion of non-target cells by anti-CD14/CD15/CD16/CD19/CD34/CD36/CD56/CD123/CD235a biotin-conjugated mono- clonal antibodies MicroBeads Kit (Miltenyi Biotec), with the use of a LS column. After isolation, T cells were stained with CFSE proliferation staining dye (Thermo Fisher) and resuspended in R10. Consecutively, T cells (1 Million cells/ml) were activated with Dynabeads Human T-Activator CD3/CD28 for T Cell Expansion and Activation (Thermo Fisher) in culture media, which consisted of LDG or PMN supernatants and fresh R10 in 1:1 ratio. For experiments, supernatants from individual patient LDGs or PMNs during the acute stage (11–14 days after fever onset (AFO)) and the convalescent stage (28–37 days AFO) were either assayed individually (n = 7 LDG, n = 3 PMN) or pooled at equal volumes (n = 4 acute, n = 2 convalescent). After 6 days of incubation at 37C (5% $CO_2$), the samples were analyzed by flow cytometry for CFSE expression to estimate the number of cell divisions and proliferation index, with the use of FlowJo software's proliferation platform (v10.0.7, BD).

## ELISA detection of IL-6, IL-8, calprotectin, MPO, GM-CSF and G-CSF

The following ELISA kits were purchased from R&D Systems and used according to the manufacturer's protocol (human IL-6 DuoSet, catalog no. DY206; human IL-8/CXCL8 DuoSet, catalog no. DY208; human calprotectin/S100A8 DuoSet, catalog no. DY4570; human MPO DuoSet, catalog no. DY3174, human GM-CSF DuoSet, catalog no. DY215; human G-CSF DuoSet ELISA, catalog no. DY214). Serum samples were diluted (1:2 for IL-6, IL-8, GM-CSF, and G-CSF or 1:1000 for MPO and calprotectin) and the concentrations of the respective analytes were determined using the standards supplied by the manufacturer. The absolute concentrations in serum were established by multiplying the determined values by the respective dilution factors.

## NET ELISA

The presence of NETs in circulation was assessed by an ELISA assay as described previously [55–57], in which 5 µg/ml mouse anti-human MPO (Bio-Rad) was coated on the surface of a Maxisorp 96-well plate overnight at +4˚C. After blocking with an incubation buffer (1% BSA in PBS), wells were incubated with 1:1000 of patient or healthy control serum for 2h and washed four times in PBS supplemented with 0,05%Tween 20 (PBS-T). Wells were then incubated with 1:50 peroxidase-conjugated anti-DNA antibody (from Cell Death detection ELISA kit, Thermo) for 90 min in incubation buffer, followed by detection with 2,2'-Azinobis [3-ethylbenzothiazoline-6-sulfonic acid]-diammonium salt (ABTS), according to manufacturer's instructions (Cell Death detection ELISA).

## Statistical analysis

Statistical analysis was performed using GraphPad Prism 8.3 software (GraphPad Software, San Diego, CA, USA) and R software v3.6.3 (R core team). Statistically significant correlations between parameters were assessed by calculating Spearman's correlation coefficients. Statistically significant differences between groups were assessed with Mann-Whitney, Kruskall-Wallis or ordinary one-way or 2-way ANOVA analyses, depending on sample distribution and the number of groups analyzed. In addition, Friedman test was used to assess significance between matched groups.

## Supporting information

**S1 Table. Characteristics of COVID-19 patients and healthy controls.**
(XLSX)

**S2 Table. Flurochrome-conjugated monoclonal antibodies used in the flow cytometry panels.**
(XLSX)

**S1 Fig. Variable peripheral blood mononuclear cell population densities in acute COVID-19.** Fresh PBMCs obtained from acute mild (outpatients) and severe (hospitalized) COVID-19 patients as well as HC were stained with a panel of antibodies (see S2 Table for details) and analyzed by flow cytometry. The whole PBMC fractions were dimensionally reduced by UMAP and clustered by FlowSOM.
(TIF)

**S2 Fig. Surface marker expression of LDG subsets in different patient groups.** The CD66$^+$CD15$^+$ LDGs formed four different clusters with variable density as identified by UMAP and FlowSOM analysis in HC, mild and severe COVID-19. A detailed heatmap

indicating mean MFI of CD33, CD11b, CD16, CD15, HLA-DR and CD66 in each LDG subset (clusters #1-#4) of HC, mild and severe COVID-19.
(TIF)

**S3 Fig. Sustained presence of promyelocytes, myelocytes and metamyelocytes in severe COVID-19.** Kinetic analysis of various circulating factors during prolonged hospitalization of two severe COVID-19 patients (patient #1 and #2). (A) LDG counts (calculated similarly as in Fig 1) (B) LDG subset frequencies (same clusters #1-#4 as in Fig 2) (C) IL-8. (D) IL-6. (E) G-CSF. (F) GM-CSF. (G) MPO-DNA. (H) calprotectin. AFO = after fever onset.
(TIF)

**S4 Fig. Comparison between LDG subsets and normal density granulocytes (PMNs) in COVID-19.** Overlay of CD16 (A) and CD11b (C) MFI heatmaps in dimensionally reduced LDG and PMN population density plots (left) and bar graphs of MFIs of individual severe COVID-19 patients (right). Dimensional reduction was performed by UMAP using CD15, CD66, CD11b, CD16 and HLA-DR MFIs of live $CD3^-/CD14^-/CD19^-/CD56^-/CD66^+/CD15^+$ population. (C-D) Detailed UMAP localization of $CD66^+/CD15^+$ cells from four individual severe (C) and mild (D) COVID-19 patients, isolated from PBMC (LDGs) or PMN fractions.
(TIF)

## Acknowledgments

The authors thank S. Mäki and M. Utriainen for expert technical assistance.

## Author Contributions

**Conceptualization:** Tomas Strandin.

**Data curation:** Pirkka T. Pekkarinen, Maria Alander, Suvi Jokiranta, Anu Patjas, Sointu Mero, Sari H. Pakkanen, Santtu Heinonen, Anu Kantele, Olli Vapalahti, Eliisa Kekäläinen.

**Formal analysis:** Luz E. Cabrera, Kirsten H. A. Nowlan, Ngoc Anh Nguyen, Suvi Kuivanen, Sointu Mero, Sari H. Pakkanen, Tomas Strandin.

**Investigation:** Luz E. Cabrera, Tomas Strandin.

**Methodology:** Pirkka T. Pekkarinen, Maria Alander, Santtu Heinonen, Anu Kantele, Olli Vapalahti, Eliisa Kekäläinen, Tomas Strandin.

**Project administration:** Anu Kantele, Olli Vapalahti, Eliisa Kekäläinen, Tomas Strandin.

**Resources:** Tomas Strandin.

**Software:** Luz E. Cabrera, Tomas Strandin.

**Supervision:** Olli Vapalahti, Eliisa Kekäläinen, Tomas Strandin.

**Validation:** Eliisa Kekäläinen, Tomas Strandin.

**Visualization:** Luz E. Cabrera, Tomas Strandin.

**Writing – original draft:** Luz E. Cabrera, Tomas Strandin.

**Writing – review & editing:** Luz E. Cabrera, Pirkka T. Pekkarinen, Maria Alander, Kirsten H. A. Nowlan, Ngoc Anh Nguyen, Suvi Jokiranta, Suvi Kuivanen, Anu Patjas, Sointu Mero, Sari H. Pakkanen, Santtu Heinonen, Anu Kantele, Olli Vapalahti, Eliisa Kekäläinen, Tomas Strandin.

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
