## [Decision Letter · Decision Letter 0]

27 May 2021

Dear Ms Cabrera Lara,

Thank you very much for submitting your manuscript "Characterization of Low-density granulocytes in COVID-19" for consideration at PLOS Pathogens. As with all papers reviewed by the journal, your manuscript was reviewed by members of the editorial board and by several independent reviewers. The reviewers appreciated the attention to an important topic. Based on the reviews, we are likely to accept this manuscript for publication, providing that you modify the manuscript according to the review recommendations.

The Reviews were split. One recommended Accept and the other recommended Rejection in large part because they felt you still did not temper your conclusions enough and acknowledge the limitations of your study and terminology. I am willing to rule in favor of publication but before that you will need to edit the paper to address the major considerations raised by the reviewer. See my requests below and also prepare a rebuttal to remainder of his/her comments.

1. You agree on the fact that LDG may not comprise of a separate phenotype of neutrophils, yet you persist with implying that LDG are such a population in the Abstract and Discussion. You also neglect to cite studies that do not support the main hypothesis that neutrophils are important in the initiation of disease up to the moment patients have to go to the ICU. **These points need to be addressed**

2. Discussion about cause and consequence. Neutrophils might still guilty by association. **This is not carefully addressed in the Discussion. **

3. A left shift during inflammatory conditions associated with the occurrence of progenitors in the peripheral blood is expected. This is normal physiology and not necessarily pathophysiology **and accordingly should be acknowledged.**

4. The complex multidimensional data supporting the presence of progenitors in the peripheral blood is not much better than the 2D plots on the expression of CD11b and CD16. You do not acknowledge Hidalgo's paper in Trends in Immunology (ref 26) that already used this gating strategy. **This acknowledgement should be clearly made.**

Sincerely,

Michael S. Diamond

Section Editor

PLOS Pathogens

Michael Diamond

Section Editor

PLOS Pathogens

Kasturi Haldar

Editor-in-Chief

PLOS Pathogens

orcid.org/0000-0001-5065-158X

Michael Malim

Editor-in-Chief

PLOS Pathogens

orcid.org/0000-0002-7699-2064

Reviewer Comments (if any, and for reference):

Reviewer's Responses to Questions

**Part I - Summary**

Reviewer #1: The role of neutrophils and neutrophil phenotypes in COVID-19 is timely and

important. The authors nicely show that the neutrophil compartment in COVID19 is different when

compared with age and sex matched controls. Unfortunately, the main take-home message is basically that acute COVID19 is associated with a left shift is not really advancing the field

Reviewer #2: This is a much improved manuscript with increased sample sizes and further detailed characterization of neutrophils in COVID-19.

**Part II – Major Issues: Key Experiments Required for Acceptance**

Reviewer #1: The article has in essence not changed as the 'main limitations' to this study has not been adequately addressed.:

1. The suggestion that LDGs belong to a separate population of neutrophils is not based on hard evidence. The work of Kaplan (who first described this concept in detail) has never put forward good scientific evidence for LDG's as seperate phenotype. The publication by Hassani et al (20) in fact suggests that the shift in density is mainly caused by activation of all phenotypes and that the density 1.077 g/ml (density of Ficoll) is completely arbitrary.

Answer: We fully agree with the reviewer that LDGs cannot be defined as a clearly different

population from “normal-density PMNs”, but rather represent distinct granulocyte subsets which are most likely also present in the PMN fraction but harbor lower density for various reasons. The evidence by us and others point out that there are at least two reasons for granulocytes to displaythe low-density phenotype: the main reason from our point of view is granulocyte immaturity (less developed granulocytes are less dense and mononuclear) and the other, as the reviewer also points out, is granulocyte activation that results in cells to lose their “normal” density. Thus, the increased presence of LDGs during a disease reflects an increased granulocytic activation and, subsequently, a replenishment of the decreased granulocyte pools, by an increased early release of immature granulocytes from the bone marrow, as demonstrated by Van Grinsven et al (2019)1

.

Not directly linked to this comment, but we have now analyzed the data also through unsupervised clustering as

a more unbiased way of identifying the different LDG subsets.

Response reviewer: although the authors "fully agree' on the issue that LDG'S are not a specific subset but either young and/or activated neutrophils, they persist throughout the article with the suggestion that LDG comprise of separate subsets of neutrophils (1-4) while the data merely show a left-shift. The choice of sorting on CD11b/CD16 and finding (pro)myelocytes, meta's and mature cells is not new and shown before in ref. 26 Hidalgo et al. This should be acknowledged.

1 Van Grinsven, E., Textor, J., Hustin, L. S., Wolf, K., Koenderman, L., & Vrisekoop, N. (2019). Immature neutrophils released in acute inflammation exhibit efficient migration despite incomplete segmentation of the nucleus. The Journal of Immunology, 202(1), 207-217.

2. The study is basically a case control study where normal matched volunteers are the control. This can lead to misinterpretation of COVID19 specific findings/mechanisms. This is important as most of the findings in the article fit with the well-known left shift generally seen in many acute diseases such as caused by infections by many micro-organisms. So a case-control study should have been performed with acute diseases other than COVID-19. It is to be expected that most of the data of the current study are also found in other infectious diseases.

Answer: We fully agree with the reviewer. As explained above, the increased circulating LDG counts most likely reflect increased egress from the bone marrow and granulocyte activation. As such, it is likely that their frequencies are increased also in any other disease with pronounced granulocyte involvement. Therefore, we have rewritten our manuscript in a way which does not emphasize the role of LDGs as a hallmark of COVID-19 specifically, but we rather aimed to characterize these cellsbetter to increase our understanding of LDGs in general.As said, it is likely that LDGs do appear also in other acute microbial infections involving strong granulocyte activation. However, considering the difficulties of assessing LDGs from fresh blood samples of acutely ill and infected patients, its time- and resource-wise not in our reach to analyze such patient samples as a reference group for our current study, unfortunately.

Response reviewer: the issue of the difficulties obtaining relevant disease controls is well taken, but should be better discussed in terms of left shifts in other acute infectious diseases.

3. The division between ICU and non-ICU is very artificial as most clinical reasons for ICU

admittance are based on clinical confounders other than COVID-19 such as cardiovascular risk factors and coagulopathy.

Answer: This is a good point and we decided to remove the ICU classification from the resubmission. We are instead currently assessing disease severity based on several different factors such as extent of oxygen supplementation and length of hospital stay. While all these parameters have their caveats, assessing several of them at the same time at least gives a better overall picture of disease severity.

Response reviewer: okay.

4. It is really a missed chance not to include/discuss the major risk factors of critical disease that are not necessarily immune driven. An important part of critical disease is caused by tissue (lung) edema, thrombo-embolische complications and coagulopathy. These are all not necessarily mediated by immune mechanisms, but much more mediated by the bradykinin system.

Answer: Yes, admittedly the detailed molecular mechanisms of COVID-19 pathophysiology are still to large extent unresolved and definitely other factors than those directly immune-related play a role. We have rewritten also the discussion part of this manuscript with this in mind. However, since we are not directly studying coagulation or bradykinin system, we feel going too deeply into these phenomena is out of the scope of this manuscript.

Response reviewer: the issue here was that the alternative pathogenesis of COVID19 might drive the disease including causing tissue damage through thrombo-embolisms and reperfusion injury. More severe disease would then lead to more release of DAMP's leading to more activation of the neutrophil compartment. So the left shift can be the consequence of disease rather than the cause. In fact, Spijkerman et al. (JLB 2021) supports this hypothesis by not finding much indications of an activated neutrophil compartment in COVID19 patients during hospital admission. This study is ignored but at least the concept should be discussed.

5. The data in the article do not support the last and essential conclusion in the abstract. The

data do not show that LDG's are major players in COVID19 pathogenesis.

Answer: This is true. Our results do not impeccably show that LDGs are major players in COVID-19 pathogenesis. We have rewritten the last sentence in the abstract as follows: “Taken together, our data confirms a significant granulocyte activation during COVID-19 and suggest a role for LDGs in COVID-19 pathogenesis.”

Response reviewer: See issue 4: the data in the literature not all support the conceprt of activation of granulocytes in COVID19 at hospital admission (initiation of disease). It is important to establish whether activation of neutrophils is associated with the cause or consequence of disease. The issue of guilty by association rather than causation is still not adequately addressed.

Reviewer #2: Overall, the authors have done a nice job of responding to Reviewer concerns. The manuscript is more comprehensive and rigorous.

**Part III – Minor Issues: Editorial and Data Presentation Modifications**

Reviewer #1: (No Response)

Reviewer #2: The figures are low-quality in this submission.

PLOS authors have the option to publish the peer review history of their article (what does this mean?). If published, this will include your full peer review and any attached files.

Reviewer #1: No

Reviewer #2: **Yes: **Betsy J Barnes

Figure Files:

Data Requirements:

Reproducibility:

References:

---

## [Editor Report · Decision Letter 1]

17 Jun 2021

Dear Ms Cabrera Lara,

We are pleased to inform you that your manuscript 'Characterization of Low-density granulocytes in COVID-19' has been provisionally accepted for publication in PLOS Pathogens.

Best regards,

Michael S. Diamond

Section Editor

PLOS Pathogens

Michael Diamond

Section Editor

PLOS Pathogens

Kasturi Haldar

Editor-in-Chief

PLOS Pathogens

orcid.org/0000-0001-5065-158X

Michael Malim

Editor-in-Chief

PLOS Pathogens

orcid.org/0000-0002-7699-2064
---

## [Editor Report · Acceptance letter]

1 Jul 2021

Dear Ms Cabrera,

We are delighted to inform you that your manuscript, "Characterization of Low-density granulocytes in COVID-19," has been formally accepted for publication in PLOS Pathogens.

Best regards,

Kasturi Haldar

Editor-in-Chief

PLOS Pathogens

orcid.org/0000-0001-5065-158X

Michael Malim

Editor-in-Chief

PLOS Pathogens

orcid.org/0000-0002-7699-2064